# Antitumor Effects of Microencapsulated *Gratiola officinalis* Extract on Breast Carcinoma and Human Cervical Cancer Cells In Vitro

**DOI:** 10.3390/ma16041470

**Published:** 2023-02-09

**Authors:** Nikita Navolokin, Maria Lomova, Alla Bucharskaya, Olga Godage, Natalya Polukonova, Alexander Shirokov, Vyacheslav Grinev, Galina Maslyakova

**Affiliations:** 1Center for Collective Use of Experimental Oncology, Saratov State Medical University n.a. V.I. Razumovsky, Saratov 410012, Russia; 2Science Medical Centre, Saratov State University, Saratov 410012, Russia; 3Laser Molecular Imaging and Machine Learning Laboratory, Tomsk State University, Tomsk 634050, Russia; 4Institute of Biochemistry and Physiology of Plants and Microorganisms, Russian Academy of Sciences (IBPPM RAS), Saratov 410028, Russia

**Keywords:** antitumor activity, cytostatic activity, polyelectrolyte microcapsules, flavonoids, extract of *Gratiola officinalis* L.

## Abstract

Flavonoid-containing *Gratiola officinalis* extract has been studied in relation to breast carcinoma and human cervical cancer cells in encapsulated and native form. Encapsulation was realized in polymer shells, which were formed by the layer-by-layer method using sequential adsorption of poly(allylamine hydrochloride) and poly(sodium 4-styrenesulfonate) on the destructible cores. The extract was prepared by the author’s method and characterized using high performance liquid chromatography. By means of optical and fluorescent microscopy, cell changes under the action of pure and encapsulated extracts were comprehensively studied, and statistical analysis was carried out. Cells were stained with propidium iodide, acridine orange, and Hoechst 33258. A fluorescence microscope with a digital video camera were used for cell imaging. The encapsulated extract caused 100% death of breast cancer SKBR-3 cells and 34% death of cervical cancer HeLa cells and prevented the formation of autophagosomes in both cultures. Analysis of the viability and morphological features of tumor cells under the action of microencapsulated extract allows us to consider microencapsulation as an effective strategy for delivering *Gratiola officinalis* extract to tumor cells and a promising way to overcome the protective autophagy.

## 1. Introduction

Encapsulation of drugs that cause a significant toxic effect on the body is one of the strategies for obtaining new pharmacological forms of drugs with little or no side effects. This approach has proven successful for a number of drugs that are already commercially available. For cancer therapy, encapsulation of toxic drugs is becoming a promising prospect for effective treatment [1].

The usage of polymeric capsules for optical tissue biopsy, simultaneously providing a therapeutic effect, has prospects both in vivo and in vitro. As is known, the role of agents for optical tissue biopsy is to amplify the fluorescent signal during biochemical reactions in tissues or cells. It is possible to control the intensity of certain processes by the increase or decrease of signals [2]. For example, the co-polymer conjugated Cy 5.5 dye is used to visualize the work of enzymes [3]. It should be taken into account that the dyes have a low molecular weight, as well as a high ability to stain any proteins. Polymeric microcapsules have in their shell, for example, as in our case, a protein with a conjugated dye. Upon ingestion and under the action of enzymes (as described in the introduction), the capsules decompose, and the free dye can be used to enhance the optical signal in vivo. In the study of Kalenichenko et al. (2021), homogeneous microcapsules containing doxorubicin (DOX) as a model anticancer drug and fluorescent semiconductor nanocrystals (quantum dots, QDs) as fluorescent nanolabels were developed. In this study, polyelectrolyte MCs provide prolonged release of DOX under conditions close to normal; tumor tissues have bright fluorescence, which opens the way to their use for tumor diagnostics and delivery of antitumor drugs [4].

Cancer is the leading cause of death worldwide, with 19.3 million new cancer cases (18.1 million excluding nonmelanoma skin cancer) and almost 10.0 million cancer deaths (9.9 million excluding nonmelanoma skin cancer) occurring in 2020 [5]. The pronounced toxicity of multicourse chemotherapy [6,7] and the formation of multiple drug resistance [8] necessitates the search for a new generation of anticancer agents and methods for their delivery, precluding the development of necrosis and creating another path of programmed tumor cell death [9,10].

Bioflavonoids exert a wide range of anticancer effects; a number of studies have established their ability to enhance the efficacy of cytostatic therapy by weakening its toxic effects on healthy cells [11,12]. Mechanisms of antitumor action of flavonoids have been established, including apoptosis induction [13,14,15,16,17,18], cell cycle arrest in the G1 or G2/M phase [19,20], inhibition of enzymes involved in metabolism, namely cytochrome P450 [21], inhibition of reactive oxygen species formation by activation of phase II metabolizing enzymes [22,23], and inhibition of angiogenesis mediated by vascular endothelial growth factor (VEGF) and basic fibroblast growth factor (bFGF) [24,25]. However, many factors can affect the bioavaibility of flavonoids. To overcome these problems, encapsulation technologies can be used target the delivery and release of bioflavonoids. As a new pharmacological form for targeted delivery, polyelectrolyte microcapsules can be applied as microcontainers for plant extracts [26,27].

The most common form for encapsulation is polymer microcapsules—polymer shells obtained by layer-by-layer adsorption of polyelectrolytes, proteins, enzymes, nanoparticles, etc., to encapsulate anticancer drugs. The size of such shells can be from hundreds of nm to units of microns [28].

Drug delivery vehicles based on polymeric microcapsules have formed the basis of scientific works of many groups due to a number of advantages: the possibility of their complete enzymatic degradation [27,29], size reduction [30], and the ability to control drug release (from rapid release to depot systems) [27].

Previously, we investigated polyelectrolyte microcapsules loaded with magnetite nanoparticles for diagnosis and therapy in rats with transplanted tumors [31].The expansion of the use of polymer carriers for encapsulation of plant components is explained by the solubility of extracts in water, where it is quite easy to form microcapsules with a polymer shell.

We demonstrated the antitumor activity of *Gratiola officinalis* extract, obtained by the author’s method [32], on human tumor cell cultures for kidney cancer (Caki-1 and SN12c), cervical carcinoma HeLa, T-cell lymphoblastic leukemia Jurkat, breast adenocarcinoma MCF-7, lung carcinoma A549, prostate carcinoma PC-3, colon carcinoma HCT-11, renal carcinoma A498, and human breast carcinoma SK-BR-3 [33,34] in vitro and on transplanted tumors of laboratory animals [35] with additional apoptotic activity in vivo. A fundamentally new approach in this study is that, for the treatment, we use *Gratiola officinalis* extract encapsulated in microcapsules. This will allow a comparative analysis of antitumor activity of encapsulated *Gratiola officinalis* extract with its pure analogue.

Thus, changes in the morphology of cell lines and their survival of human breast carcinoma SK-BR-3 and cervical cancer HeLa cells will be analyzed microscopically and using standard biological research methods under the action of *Gratiola officinalis* L. extract encapsulated in polymer shells. Quantitative characteristics obtained by processing optical fluorescent images of cells under encapsulated extracts will help to describe the nature of cell destruction.

## 2. Materials and Methods

### 2.1. Materials for Microcapsule Preparation

Calcium chloride, sodium carbonate, poly(allylamine hydrochloride) (PAH), poly(sodium 4-styrenesulfonate) (PSS), bovine serum albumin, tetramethylrhodamine fluorescent dye (TRITC), dextran sulfate sodium salt (DS), and ethylene diamine tetraacetic acid disodium salt were purchased from Sigma Aldrich (Germany) and used as received; they were ACS grade. Milli-Q water (18.2 MΩ·cm) was used in all experiments.

### 2.2. Gratiola officinalis L. Extract Preparation and Description of the Chemical Composition

*Gratiola officinalis* L. extract was obtained according to the original author’s method [32]. The voucher specimen *Gratiola officinalis* L. is in the Biocollection of Herbarium Specimens of Saratov State Medical University n.a. V.I. Razumovsky, Family —Plantaginaceae, Genus—Gratiola, Species—*Gratiola officinalis* L. [36]. The investigated composition of biologically active substances (BAS) from *Gratiola officinalis* L. wasa dried extract, yellow-brown in color, mixed with water and ethyl alcohol in any ratio.

The chemical composition of *Gratiola officinalis* extract was investigated by high performance liquid chromatography--tandem mass spectrometry(HPLC-MSMS [37]. HPLC analysis of the extract was done using the Dionex Ultimate 3000 chromatograph (Thermo Fisher Scientific, Waltham, MA, USA) coupled with a Macherey-Nagel NucleodurHTecC18 analytic column, with an average particle diameter of 5 µm, pores of 100 Å, and a geometry of 150 × 3.0 mm. Flow rate was 0.25 mL/min, and total analysis time was 25 min. Detection was performed at wavelengths of 342, 360, and 320 nm using a Dionex™ UltiMate™ DAD 3000 1024-element diode-array detector (Thermo Fisher Scientific, Waltham, USA); the 3Dfield in the 200–400 nm range was also registered for the analysis of electronic spectra of the components. Chromatographic separation of the extract was performed under conditions of gradient elution (Solvent A—acetonitrile of HPLC grade (Panreac Química SLU, Barcelona, Spain), Solvent B—trifluoroacetic acid (TFA, Sigma Aldrich, St. Louis, MO, USA)) solution (pH 2.5): the composition of the mobile phase was changed as follows: 0–10 min: 15% A, 85% B; 10–19 min: 15→70% A, 85→30% B; 19–20 min: 70% A, 30% B; 20–22 min: 70→15% A, 30→85% B; 22–25 min: 15% A, 85% B. Total flow was 2 mL/min. Injection volume of the sample was 50 µL. Detection was performed at wavelengths of 342, 360, and 320 nm. Chromatograph control and data analysis were performed by Chromeleon version 7.1.2.1478 (Thermo FisherScientific, Dionex, Waltham, MA, USA). All collected fractions were concentrated on the rotary evaporator Laborota 4000 (Heidolph, Schwabach, Germany) followed by lyophilization using Benchtop 2K (VirTis, Hampton, New York, NY, USA).

The following compounds a polyphenolic nature were detected in the chemical composition of the extract, which were structurally identified by the built-in library of plant metabolites as 7,3′-di-O-luteolin glycoside (r.t. 4.46 min, *m*/*z* 285.0431, *m*/*z* of reference compound from DB, 285.0396), apigenin 7-O-glucoside (apigetrin, r.t. 4.93 min, *m*/*z* 431.0979, *m*/*z* of reference compound from DB, 431.0958), as well as in trace amounts of eupatilin (r.t. 8.01 min, *m*/*z* 313.0360, *m*/*z* of reference compound from DB, 313.0358). In addition, 3-(1-2)-glucoside (glycoside or galactoside) of soyaspogenol B (soyasapogenol B, *m*/*z* 795.4532), which is structurally a pentacyclic triterpenoid, was found [37].

### 2.3. Polyelectrolyte Microcapsules Preparation Method

Encapsulation of the extract in polyelectrolyte shells was performed using the method of sequential adsorption of polyelectrolytes onto the surface of a subsequently extracted solid porous spherical core as first described in G.B. Sukhorukov [38]. *Gratiola officinalis* L. extract was deposited in the pores of spherical calcium-carbonate microparticles with an average size of 3–5 μm by the coprecipitation method [39]. Polyelectrolyte shells on the cores were produced by layer-by-layer adsorption of molecules on spherical cores during their mixing in water solutions (concentrations of DS, PAH and PSS were 2 mg/mL). Bovine serum albumin (BSA) was labeled with tetramethylrhodamine (TRITC) fluorescent dye using the method described in [40]; the concentration of BSA-TRITC water solution was 2 mg/mL. Final shell composition was PSS/PAH/BSA-TRITC/PAH/PSS/PAH/DS (bovine serum albumin labeled with tetramethylrhodamine fluorescent dye (BSA-TRITC), dextran sulfate sodium salt (DS), poly(allylamine hydrochloride) (PAH), poly(sodium 4-styrenesulfonate) (PSS)). The concentration of *Gratiola officinalis* extract in the capsule suspension was 0.18 mg/mL, determined by spectrophotometric measurements using a spectrophotometer Synergy H1 (Bio Tek, Santa Clara, CA, USA). The concentration of the encapsulated extract was measured by analyzing the supernatants that were obtained during the preparation of the microcapsules. The microcapsule preparation process is a layer-by-layer deposition of polymers on solid cores; after each adsorption step, the carrier suspension is washed with water, resulting in the formation of a supernatant. Using the calibration curve of the extract and analyzing the absorption spectrum of the supernatants at a wavelength of 520 nm, it was determined that 1 mL of the microcapsule suspension contains 0.18 mg of the extract; the measurement error was no more than 5%,and the limit of detection of the spectrophotometer Synergy H1 (Bio Tek, Santa Clara, CA, USA) was <1% at 3.0 OD. The last layer of dextran sulfate was necessary to reduce microcapsule aggregation. Previously, it was shown that it is dextran that helps to keep microcapsules in the form of a suspension for a sufficiently long time without precipitation, while at the same time imparting a negative charge on the surface of microcapsules [41].

Images of microcapsules with and without *Gratiola officinalis* extract was obtained by scanning electron microscopy (MIRA II LMU scanning electron microscope, Tescan, Brno, Czech Republic; the operating voltage was 30 kV, in secondary electron mode, and samples were covered by gold before measurements) and is shown in Figure 1. Figure 1 shows that the capsules with the extract have slight damage in the shell, which we connect to the precipitation of extract microcrystals when the capsules are dry. When the extract is in an aqueous solution, no precipitation of crystals occurs; otherwise, the formation of microcapsule shells would be impossible.

### 2.4. Cultivation of SK-BR-3 and HeLa Cell Lines

The cell cultures of human breast carcinoma SK-BR-3 and cervical cancer HeLa cells were obtained from the cell culture collection of the National Medical Research Center of Oncology n.a. N.N. Blokhin, Moscow, Russia. HeLa cervical cancer cells were cultured in plastic vials (Corning, Biologix, Jinan, China) in RPMI medium 1640 (PanEco, Russia), with the addition of 10% fetal bovine serum (FBS, Millipore, Sigma-Aldrich, Supelco), in a CO_2_ incubator at 37°C in an atmosphere with 5% CO_2_. For the experiment, cells were grown in a 24-well flat-bottomed culture plate (Corning, Biologix, China) for 24 h until 70–80% monolayer formation. Then, pure or encapsulated extract at a concentration of 0.18 mg/mL was added to the culture, depending on the experiment, and incubated for 24 h. HeLa cells were then stained with two dyes simultaneously: propidium iodide, which penetrates non-viable cells, destroying their membrane and staining cells red, and acridine orange, which stains living cells green.

The human breast carcinoma SK-BR-3 cells were cultured on complete nutrient medium RPMI-1640 (PanEco, Moscow, Russia) with the addition of 10% fetal bovine serum (TES, HyClone, USA), 2 mM/mL glutamine (PanEco, Moscow, Russia), and 50 mg/mL penicillin-streptomycin (PanEco, Moscow, Russia) at 37 °C in an atmosphere with 5% CO_2_. Cells were grown on slides up to a 70–80% monolayer. The effect of pure *Gratiola officinalis* extract at 0.18 mg/mL and 0.9 mg/mL and the effect of empty microcapsules and microcapsules containing *Gratiola officinalis* extract at 0.18 mg/mL were studied on each SK-BR-3 cell culture. The control was SK-BR-3 cell culture without any effect.

After incubation for 24 h, cells were fixed in alcohol and acetone. Then, they were stained with Hoechst 33258 dye (1 µg/mL, PanEco, Moscow, Russia). Cells were placed under coverslips using Fluorescent mounting medium (Dako, Glostrup, Denmark). The samples were analyzed using a Nikon 80i fluorescence microscope (Nikon Group Companies, Tokyo, Japan) at 435–485 nm; Hoechst 33258 bound to the structures of the cell nucleus.

### 2.5. Optical and Fluorescence Microscopy Statistical Analysis

A Leica DMI fluorescence microscope was used to visualize the cells. Images were captured and analyzed using a Leica DFC 420 C digital video camera (Leica Microsystems, Heerbrugg, Switzerland) and Leica Application Suite Software Version: LAS V3.1 (Leica Microsystems, Heerbrugg, Switzerland). ImageJ software (Java-based image analysis public domain software) was used to count the number of cells.

The use of two fluorescent dyes, acridine orange (excitation peak at 490 nm and emission peak at 520 nm) and propidium iodide (excitation peak at 535 nm and emission peak at 615 nm) in the “living and dead” test allowed detection of the number of necrotic cells (nucleus stained red), viable cells (cells stained green), cells with autophagosomes (cells with black “vacuoles” in the cell), cells with apoptosis (nucleus sharply condensed and stained bright green-yellow), and apoptotic corpuscles (stained light green).The “alive and dead” test allowed us to assess the stage of pycnosis of the nucleus, which was present in apoptosis if the nucleus was stained bright green. Parallel examination of cells using phase contrast microscopy was necessary to confirm the presence of nuclei in the karyoplasm and autophagosomes and other formations in the cytoplasm.

The following parameters of HeLa cells were evaluated after acridine orange and propidium iodide staining: total number of cells, number of living and number of dead cells per field of view. Then, we calculated the percentage of living and dead cells, as well as the proportion of cells in apoptosis (pycnosis, apoptotic bodies, “sickles”) and the proportion of cells with autophagolysosomes from the number of living cells per field of view.

The following parameters of SK-BR-3cells were analyzed after Hoechst 33258 staining: average number of cells in the field of view; culture growth activity—the ratio of the average number of all cells in the field of view after treatment to the average number of all cells in the field of view in control samples; absolute number of mitoses in the field of view; mitotic activity index—ratio of the number of cells in mitotic stages to the total number of cells in the field of view, multiplied by 100; absolute number of apoptosis in the field of view; apoptotic activity index—ratio of numbers of cells in apoptosis to the total number of cells. The cells in each group were counted in at least ten fields of view at magnification ×200.

The statistical analysis was performed using Microsoft Office Excel software. Normality of distribution of indices in the groups was checked using the Shapiro–Wilk criterion. For parametric distribution in the studied groups the arithmetic mean (M) and standard deviation (δ) were calculated. To compare the indices obtained in the study with their parametric distribution, but without equality of variance, the Cramer-Welch test (T) was used, in which the difference of arithmetic mean of the two samples (control and experimental) was divided by the natural estimate of the mean square deviation of this difference. This method allows for multiple comparisons across groups [42]. In this method, differences in the mean with a probability of more than 95% (*p* > 0.05) were determined when T ≥ 1.96.

## 3. Results

### 3.1. Effect of the Encapsulated Extract Gratiola officinalis L. on Breast Carcinoma Cells (SK-BR-3)

After 24 h in SK-BR-3 cell culture without any treatment, the average number of living cells was 365 cells per field of view, of which 0.6% were in metaphase, anaphase and telophase stages of mitosis and 0.56% of cells had apoptotic signs. At the same time, the chromatin of cell nuclei was not intensely stained (Figure 2a). Twenty-four hours after incubation of SK-BR-3 cells with empty microcapsules, no change in the number of cells was observed in comparison with the control group.

The average number of living cells and the number of cells in mitosis did not differ from the control after treatment by *Gratiola officinalis* extract at a concentration of 0.18 mg/mL. The number of cells with signs of apoptosis increased 1.6 times. Most of them were detected as apoptotic corpuscles (75%); the remaining cells had nucleus pyknosis (25%). After treatment with *Gratiola officinalis* extract at a concentration of 0.9 mg/mL (Figure 2b), we registered a 35% reduction of living cells in comparison with the control and an absence of cells at mitosis stages (Table 1), testifying to the presence of the cytostatic activity of the extract. The vast majority of cells had clear signs of apoptosis (86%), namely condensed, brightly stained nuclear chromatin (Figure 2c), indicating the presence of marked apoptotic activity of extract.

All cells of the SK-BR-3 culture were already dead and destroyed 24 h after treatment by encapsulated *Gratiola officinalis* extract (Figure 2d); cell debris with undestroyed capsules of round shape was noted. Such a pronounced effect was not observed even at a higher concentration (0.9 mg/mL) of *Gratiola officinalis* extract without capsules (Figure 2b, Table 1).

### 3.2. Effect of the Encapsulated Extract Gratiola officinalis L. on Human Cervical Cancer (HeLa) Cells 

Twenty-four hours after seeding and incubation, HeLa cell culture without any treatment was represented by an even monolayer of cells, tightly adhering to each other and occupying most of the surface (Figure 3). The cells were mostly attached to the substrate; the cells were spindle-shaped, pear-shaped, or polygonal. There were single cells of the syncytium or symplast type. The cells had a homogeneous cytoplasm and distinct nucleus; after staining with acridine orange and iodide propidium, small orange granules of lysosomes appeared in the cell cytoplasm in the near-nuclear region. The average number of living cells per field of view was 324 ± 32.3. Dead cells were single (Figure 3).

The number of nucleolus varied from one to four; more often there were two to three, indicating active rRNA synthesis in the nucleolus-forming zones of the chromosomes. Single autophagosomes were found in single cells, not connected to lysosomes (Figure 4). One cell with two large autophagosomes of round shape and with a nucleus displaced to the periphery was found (Figure 4). No apoptotic cells were detected.

Twenty-four hours after incubation of HeLa cells with empty microcapsules, no changes in morphology or cell number were observed in comparison with the control group, but an increase of cells with autophagosomes was noted.

### 3.3. Morphological Changes after Exposure to Non-Encapsulated Gratiola officinalis Extract

Twenty-four hours after incubation with *Gratiola officinalis* extract at a concentration of 0.18 mg/mL, cells were mainly a polygonal shape, and the monolayer was fragmentarily presented (Figure 5). Activity of nucleolus organizers was preserved; the number of nucleolus in the nucleus was more often from two to three. The average number of living cells decreased two-fold compared to the control, which indicated a high cytostatic activity of *Gratiola officinalis* extract. The appearance of dead cells after incubation with *Gratiola officinalis* extract in the low concentration of 0.18 mg/mL (Table 1) indicated its significant cytotoxic activity. Most of the dead cells were at the picnosis stage, which suggested a high apoptotic activity of the extract.

We observed the appearance of autophagosomes in tumor cells under the influence of pure extract at the concentration of 0.18 mg/mL (Figure 5, Table 1). The cell agglomeration and cytoplasmic contacts were no longer formed at this concentration of *Gratiola officinalis* extract.

At extract concentrations of 0.9 mg/mL, cells became oval in shape, indicating a decrease in cell–substrate bonding and their detachment from the substrate. The cells lay in separate groups, not forming a monolayer (Figure 6). No autophagosomes were formed at the concentration of *Gratiola officinalis* extract of 0.9 mg/mL. The number of nucleolus in the nucleus was reduced to one, while their weaker staining was observed, indicating a decrease in the activity of nucleolus organizers (Figure 6). Cells with a bright luminescence of the nucleus were observed, which indicated chromatin condensation and possible initiation of apoptosis; however, there were few apoptotic corpuscles (Table 1).

Twenty-four hours after treatment by encapsulated extract of *Gratiola officinalis*, the autophagosomes in tumor cells were absent in HeLa cells, in contrast to the groups with treatment by empty microcapsules and pure extract in a comparable concentration (Figure 7). In addition, the number of dead cells (by 34%) and cells in a state of apoptosis (by 9%) increased compared to the control, and the proportion of cells with pycnosis signs decreased after treatment by capsulated *Gratiola officinalis* extract compared to the action of pure *Gratiola officinalis* extract. After treatment by pure extract, autophagy of up to 71% was observed in tumor cells, while no autophagy was observed in tumor cells under the effect of the encapsulated extract in comparable concentrations. This effect is apparently related to the degradation of microcapsules inside the cell, which provides a high concentration of the extract, prevents the formation of autophagosomes, and causes tumor cell death. It should be noted that the empty capsules did not cause cell death or change in cell number but induced autophagy in tumor cells in 54 ± 10%.

Thus, pronounced death of tumor cells and lack of cytoprotective autophagy were observed in human breast cancer SK-BR-3 and cervical cancer HeLa tumor cells after treatment by encapsulated *Gratiola officinalis* extract.

## 4. Discussion

There is a large number of publications in the literature devoted to bioflavonoids, which are the main group of polyphenols distributed in the products of plant origin. Bioflavonoids play an important role in maintaining human health and fighting numerous diseases, including cancer, diabetes, and cardiovascular, neurodegenerative, and inflammatory diseases of various organs and systems. Despite this useful biological activity, flavonoids are not stable in environmental conditions and have low solubility in water and low bioavailability after oral administration, which limits their use in pure form. We found that the extract of *Gratiola officinalis* contains a large amount of bioflavonoids and has antitumor activity [32,33,34,35]. However, we didfind in the literature data on encapsulation of *Gratiola officinalis* extract.

Encapsulation is a promising tool for preserving the useful properties of bioflavonoids and eliminating any adverse aspects of their use (bitter taste, unpleasant smell, etc.). Various methods have been used for encapsulation of flavonoids. The widely used spray-drying method is very common in the food industry because of its low cost. However, the high temperature at which the food is processed leads to the destruction of compounds such as vitamins and flavonoids [44,45].

A large number of studies have been conducted using cyclodextrins as capsule shells, which, according to the authors, improve the dissolution rate, stability, solubility, and bioavailability of bioactive ingredients [46,47,48].

It has been shown that the use of liposomes as carriers for different bioactive compounds, including flavonoids, is promising [49]. However, high production cost, low physicochemical stability, and rapid release of the main substance in the gastrointestinal tract are the main drawbacks of their application [50].

The polymeric nanoparticles are the most interesting for bioflavonoid encapsulation. They can be made using various ingredients, such as natural and synthetic polymers and inorganic materials [46,51]. Encapsulation of quercetin, including lipid-based carriers, polymer nanoparticles, inclusion complexes, micelles and conjugate-based delivery systems, were presented in a review [51].

One of the key characteristics of encapsulation systems is the lack of negative impact on flavonoid bioavailability [52]. Previously, we investigated polyelectrolyte microcapsules loaded with magnetite nanoparticles for diagnosis and therapy in rats with transplanted tumors [31]. Multilayer capsules of 4 microns in biodegradable polymers loaded with iron oxide nanoparticles were injected intravenously into rats. The distribution of the microcapsules in the organs was investigated by magnetic resonance imaging, histological examination, atomic absorption spectroscopy, and electron spin resonance. A comprehensive analysis of the biodistribution and degradation of polymeric microcapsules loaded with magnetite nanoparticles gave us grounds to recommend these composite microcapsules as effective and safe tools for diagnostic and drug delivery applications.

The results allowed us to conclude that the encapsulated form of *Gratiola officinalis* demonstrates a significant advantage over the action of the pure extract. Perhaps, the increased cytotoxic and cytostatic activity of the extract is due to the increased local concentration of the extract. There is another aspect of the application of encapsulated forms of the extract in the treatment of cancer. It is conditioned by the fact that under prolonged use of cytostatic drugs, the tumor cells develop resistance to their action, the morphological manifestation of which is the formation of autophagosomes. In our earlier studies, we obtained data confirming the ability of free extract of *Gratiola officinalis* to block the development of autophagy in tumor cells [53,54], which is consistent with the literature data about the interaction between autophagy and apoptosis [10,16,21].

## 5. Conclusions

Flavonoids, being important components of many plant products, play an important role in maintaining human health and are used to treat various diseases. The problems of their bioavailability, stability, and efficacy can be solved by the use of microencapsulation.

Our study on two tumor cell cultures (breast cancer SKBR-3 and cervical cancer HeLa) revealed a higher antitumor efficacy of the microencapsulated extract of *Gratiola officinalis* compared to the unencapsulated extract. The encapsulated extract caused 100% death of breast cancer SKBR-3 cells and 34% death of cervical cancer HeLa cells and in both cultures prevented the formation of autophagosomes.

Analysis of the viability and morphological features of tumor cells after exposure to the encapsulated extract of *Gratiola officinalis* allows us to consider the use of microencapsulated forms of *Gratiola officinalis* extract as a promising way to treat cancer and overcome protective autophagy.

Obtaining plant extracts is a promising approach in the treatment of severe diseases. Unlike complex chemical compounds, the protocols for obtaining plant extracts do not require new machine complexes, and the issue with the resource base will be resolved easily. The future development of flavonoid delivery systems with antitumor effects may be associated with the development of encapsulation methods with higher efficiency, which will make a significant contribution to the fight against cancer.

## Figures and Tables

**Figure 1 materials-16-01470-f001:**
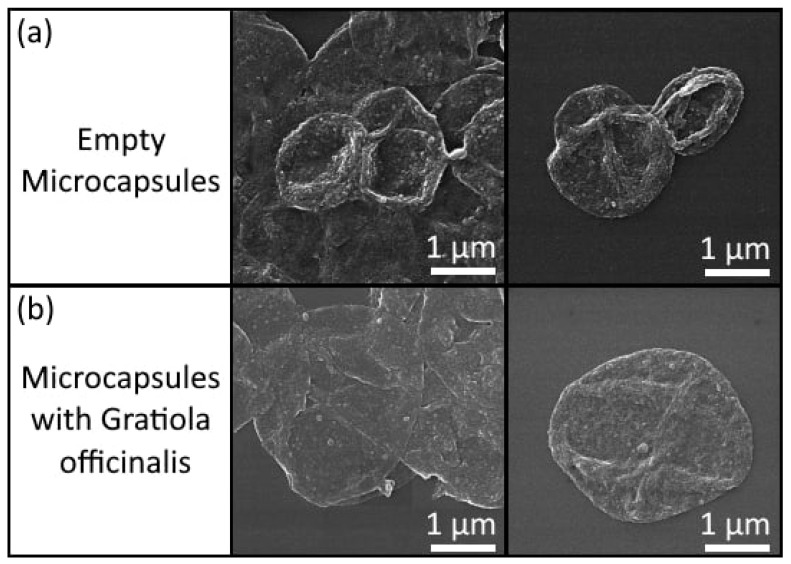
Images of PSS/PAH/BSA-TRITC/PAH/PSS/PAH/DS microcapsules, obtained by scanning electron microscopy. Images of microcapsules with (**b**) and without (**a**) *Gratiola officinalis* extract are shown at their various concentrations in solution.

**Figure 2 materials-16-01470-f002:**
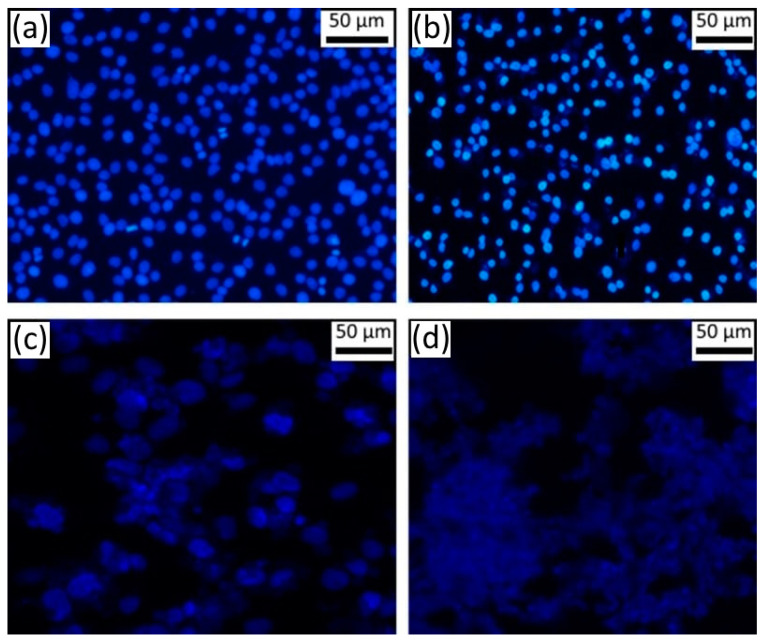
(**a**) Breast cancer cells of SK-BR-3 line without any treatment, fluorescence mode at 435–485 nm, stained with Hoechst 33258; (**b**) SK-BR-3 cells after treatment by *Gratiola officinalis* L. extract at a concentration of 0.9 mg/mL; (**c**) after treatmentby empty microcapsules; (**d**) after treatment by encapsulated *Gratiola officinalis* L. extract at a concentration of 0.18 mg/mL. Magnification ×200.

**Figure 3 materials-16-01470-f003:**
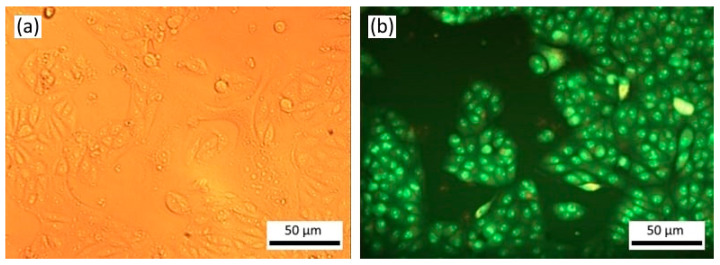
HeLa culture cells without any treatment after 24 h: (**a**) phase contrast microscopy; (**b**) fluorescence microscopy, acridine orange and propidium iodide staining. Magnification ×200.

**Figure 4 materials-16-01470-f004:**
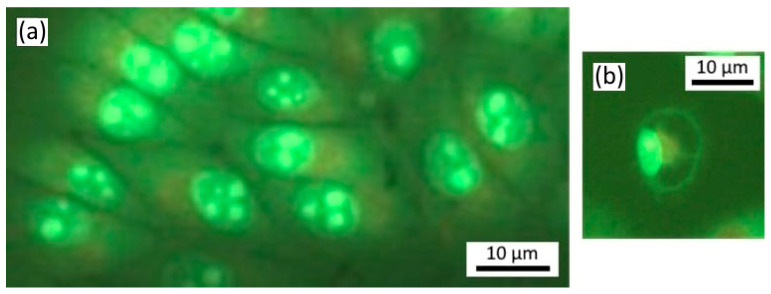
HeLa culture cells without any treatment after 24 h: (**a**) nuclei, (**b**) autophagosomes. Fluorescence microscopy with propidium iodide and acridine orange staining. Magnification ×1000.

**Figure 5 materials-16-01470-f005:**
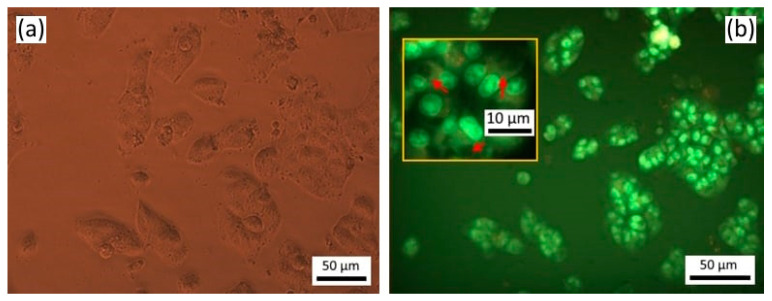
HeLa culture cells 24 h after incubation with 0.18 mg/mL *Gratiola officinalis* extract, autophagosomes: (**a**) phase contrast microscopy, (**b**) fluorescent microscopy, staining with acridine orange and propidium iodide. Magnification ×200. Arrow indicates autophagosomes. Copyright 2023 by Springer Nature and Sons. Reprinted with permission from Ref. [43].

**Figure 6 materials-16-01470-f006:**
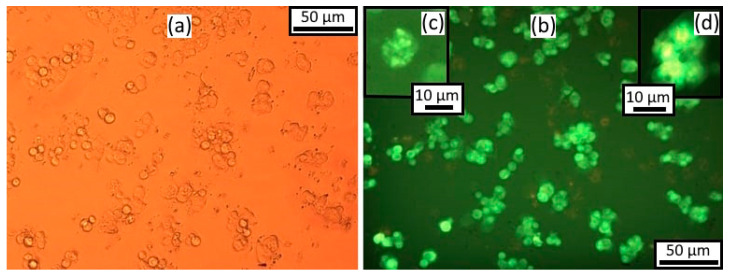
HeLa culture cells 24 h after incubation with 0.9 mg/mL *Gratiola officinalis* extract, (**a**) phase contrast microscopy; (**b**) fluorescent microscopy, staining with acridine orange and propidium iodide. Magnification ×200; (**c**) nucleus fragmentation; (**d**) autophagosomes. Magnification ×1000. Copyright 2023 by Springer Nature and Sons. Reprinted with permission from Ref. [43].

**Figure 7 materials-16-01470-f007:**
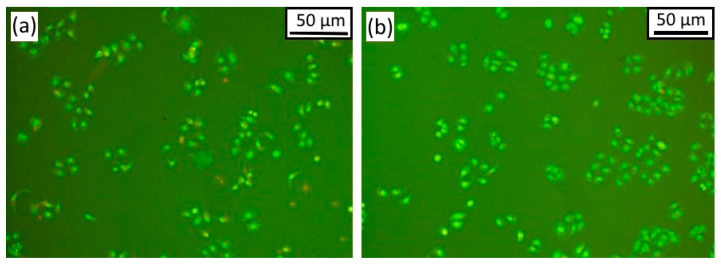
HeLa cervical cancer cell line after 24 h. Magnification 200. (**a**) After treatment by empty capsules; (**b**) after treatment by encapsulated *Gratiola officinalis* extract at a concentration of 0.18 mg/mL. Fluorescence microscopy, staining with acridine orange and propidium iodide.

**Table 1 materials-16-01470-t001:** Morphological indicators of tumor cell viability and death under *Gratiola officinalis* L. extract in pure and encapsulated forms.

Cellculture	SK-BR-3		HeLa			
Groups	Number of Viable Cells	Proportion of Cells in Apoptosis	Number of Viable Cells	Proportion of Dead Cells, %	Proportion of Cells with Pycnosis, %	Proportion of Cells with Autophagosomes, %
Control	365 ± 24.59	0.56 ± 0.3	324 ± 32.3	0 ± 0	0 ± 0	1 ± 1
Extract 0.18 mg/mL	354.6 ± 4.2	2.3 ± 0.04 *	168 ± 15 *	2.8 ± 0.5 *	61 ± 8 *	71 ± 11 *
Extract 0.9 mg/mL	236.6 ± 7.8 *	86.3 ± 1.4 *	133 ± 12 *	13.2 ± 4 *	71 ± 11 *	0.3 ± 0.1 *
Empty capsules	342 ± 11	1 ± 0.5	330 ± 40.8	0 ± 0	0 ± 0	54 ± 10 *
Encapsulated extract 0.18 mg/mL	0 ± 0 *	0 ± 0 *	180 ± 19 *	34 ± 0.5 *	9 ± 3 *	0 ± 0

Note: * *p* < 0.05; the reliability of differences between the values of the experimental and control groups was determined by Cramer–Welch test.

## Data Availability

Not applicable.

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
