# Peer review of "Antitumor Effects of Microencapsulated Gratiola officinalis Extract on Breast Carcinoma and Human Cervical Cancer Cells In Vitro"

_materials, 2023, doi:10.3390/ma16041470_

Round 1

Reviewer 1 Report

This manuscript is written very poorly and the results are demonstrated incompletely. For example, in Abstract, no one mentions about the ‘Materials and Methods’. Introduction lacks a proper and concise explanation about the uniqueness and motivation about this work. It neither gives any distinct idea as to why this study is so important as compared to what has been done previously.

As mentioned before, the manuscript is written very poorly, for example, ‘Result’ and ‘Discussion’ must be together. Also, in text, the figure is not written with its relevant numbers e.g. 194. In some figures (Figure 1), even the scale is not visible.

Analysis is not conducted well and hence incomplete. The hydrodynamic diameters and zeta-potentials of the same must be determined by dynamic light scattering (DLS). The morphology of encapsulated Gratiola officinalis in polyelectrolyte must be examined by using high resolution TEM. Chemical analysis of encapsulated Gratiola officinalis in polyelectrolyte before and after encapsulation must be done as well.

Conclusion does not reflect as to what has been done. It is very short and does not sound very promising.

Reviewer 2 Report

The manuscript by Navolokin et al. studies the "Effects of microencapsulated Gratiola officinalis extract on breast carcinoma and human cervical cancer cells in vitro." Overall, the topic is interesting, and my research group will cite, if published, to describe the change in cancer cell morphology. Well-written manuscript. There are minor corrections to consider.

Title: Authors stated in the title that they will study ”Effects of microencapsulated……”. However, they omitted what effects they were looking to study. They will study the effect on morphology and viability. Otherwise, it is not clear what they will study specifically.

Abstract: The abstract is missing the morphology change description. In addition, please add the name of the instruments used in the study for the detection of morphology change and dead cells.

Keywords: “flavonoids” is stated in the keywords but not mentioned in the abstract.

Overall, the manuscript is missing literature discussion and comparison with the current work in the introduction and all parts. Authors are recommended to include a table toward the end and compare the morphology and viability change with similar systems.

The authors could have included additional studies, but they were able to use the available techniques to address the concept. However, they can discuss other literature approaches to study the same. The submitted manuscript is missing the literature comparison to justify the novelty of this work. it is a weak part overall. Do gold NPs, polymers, and other materials have the same effect?

Labels associated with the figure are scattered all over the text. Please fix. It probably occurred while the transformation to .pdf.

I did not see, unless I missed it, the reason for choosing the Gratiola officinalis extract for this study. In addition, what are the benefits of Gratiola officinalis extract in general? What is a Gratiola officinalis extract? Any commercial comments that the authors can add? Please tell the readers some information.

The materials part is not included. Authors are recommended to state all used chemicals, purity, and source.

Figure numbers must be checked, they are not in order. The editorial office and authors must check the manuscripts before forwarding them to the reviewers. For example, what is Figure 76?

6. Patents This section is not mandatory but may be added if patents result from the work reported in this manuscript. Why is this included? 

Round 2

Reviewer 1 Report

In images, for example Figure 1, the scale is not visible. Neither the images are labeled as a) and b) and so on. 

Also, the experimental analysis for encapsulation still seems insufficient given the fact that for similar studies various techniques such as UV-VIS, FT-IR , and Fluorescence spectroscopy are used to determine the encapsulation.

Author Response

Many thanks for the constructive comments. Below we provide point-by-point responses in the following format: the the reviewer’ comments are given in (А), our comments are given in (B):

(A) In images, for example Figure 1, the scale is not visible. Neither the images are labeled as a) and b) and so on. 

(B) Thank you! All pictures were found to a consensus. Made on a), b), etc. in the parts figures panels. The scale bar is easily visible in all figures now.

(A) Also, the experimental analysis for encapsulation still seems insufficient given the fact that for similar studies various techniques such as UV-VIS, FT-IR , and Fluorescence spectroscopy are used to determine the encapsulation.

(B) The content of the extract in microcapsules was determined using UV-Vis spectroscopy. First, a calibration curve was built. During the preparation of the shells that encapsulated the extract, the supernatant was collected. The production of microcapsules is accompanied by layer-by-layer adsorption of polymers on the surface of micron particles; after each adsorbed layer, the particles are washed to remove the unreacted polymer and/or the released extract. Analysis of the supernatants, which were collected in layers, allowed us to determine the amount of encapsulated extract, given that we knew the initial amount of extract in the carriers before the formation of polymer shells. Using the calibration curve of the extract and analyzing the absorption spectrum of the supernatants at a wavelength of 520 nm, it was determined that 1 ml of the microcapsule suspension contains 0.18 mg of the extract, the measurement error was no more than 5%, limit of detection of spectrophotometer Synergy H1 (Bio Tek, USA) is <1% at 3.0 OD. In less detail, all information on determining the content of the extract in microcapsules was added to the text of the article in the section "Polyelectrolyte microcapsules preparation method".